# Learning to Plan Before Answering: Self-Teaching LLMs to Learn Abstract Plans for Problem Solving

**Jin Zhang**[1,2]**, Flood Sung**[2]**, Zhilin Yang**[2]**, Yang Gao**[1]**, Chongjie Zhang**[3]

[1]Institute for Interdisciplinary Information Sciences, Tsinghua University, China
[2]Moonshot AI
[3]Washington University in St. Louis
`jin-zhan20@mails.tsinghua.edu.cn`

## Abstract

In the field of large language model (LLM) post-training, the effectiveness of utilizing synthetic data generated by the LLM itself has been well-presented. However, a key question remains unaddressed: what essential information should such self-generated data encapsulate? Existing approaches only produce step-by-step problem solutions, and fail to capture the abstract meta-knowledge necessary for generalization across similar problems. Drawing insights from cognitive science, where humans employ high-level abstraction to simplify complex problems before delving into specifics, we introduce a novel self-training algorithm: LEarning to Plan before Answering (LEPA). LEPA trains the LLM to formulate anticipatory plans, which serve as abstract meta-knowledge for problem-solving, before engaging with the intricacies of problems. This approach not only outlines the solution generation path but also shields the LLM from the distraction of irrelevant details. During data generation, LEPA first crafts an anticipatory plan based on the problem, and then generates a solution that aligns with both the plan and the problem. LEPA refines the plan through self-reflection, aiming to acquire plans that are instrumental in yielding correct solutions. During model optimization, the LLM is trained to predict both the refined plans and the corresponding solutions. By efficiently extracting and utilizing the anticipatory plans, LEPA demonstrates remarkable superiority over conventional algorithms on various challenging natural language reasoning benchmarks.

## 1 Introduction

Large Language Models (LLMs) have revolutionized the field of natural language processing, demonstrating remarkable capabilities in handling complex language tasks (Achiam et al., 2023; Zhao et al., 2023; Yang et al., 2024; Shahriar et al., 2024). While post-training optimization of LLMs demands a substantial volume of data (Xiao et al., 2023; Wang et al., 2024b), recent works reveal that LLMs obtain the potential of generating high-quality synthetic data themselves (Zelikman et al., 2022; Gulcehre et al., 2023; Singh et al., 2023; Bansal et al., 2024). These works, known as self-training methods, improve the LLM by iterating between generating data with LLMs and optimizing LLMs with the generated data. Self-training methods alleviate the requirement of expensive human annotations and make post-training much more scalable.

A central challenge in self-training is, what essential information should such self-generated synthetic data encapsulate? Despite remarkable progress, this problem has not been well studied. Previous works only generate step-by-step problem solutions, and train the LLM to maximize the log-likelihood of generating these solutions (Zelikman et al., 2022; Singh et al., 2023). This approach only trains the LLM to memorize knowledge about task-specific solutions, and fails to capture the high-level abstract meta-knowledge necessary for generalization across similar problems. As a con-

sequence, previous self-training methods obtain only limited generalization abilities, and struggle on difficult natural language tasks such as Hendrycks MATH (Hendrycks et al., 2021).

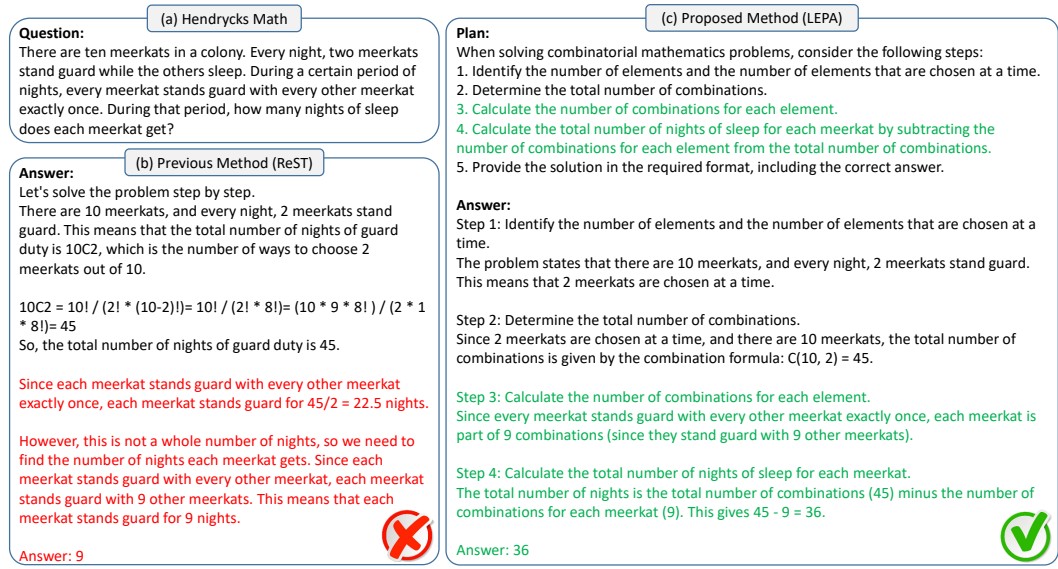

Figure 1: A didactic example demonstrating how LEPA outperforms baseline methods by learning to generate anticipatory plans before answering. (a) An example problem in the Hendrycks MATH test set. (b) An incorrect solution given by the LLM trained with a baseline method, ReST. The model fails to generate correct reasoning steps. (c) A correct solution given by the LLM trained with our proposed method, LEPA. The model generates high-quality plans, and then follows the plan to solve the problem correctly.

To tackle this challenge, we draw insights from cognitive science (Wang & Chiew, 2010; Radüntz, 2020): humans simplify complex problems through high-level abstraction before engaging with details (Ross, 2009). Such abstraction not only lightens the cognitive load but also distills high-level meta-knowledge that is transferable to analogous problems. This idea is also evidenced by recent advances in meta-learning (Finn et al., 2017; Rakelly et al., 2019), which learn generalizable meta-knowledge that enables fast adaptation to similar problems. We propose a novel self-training algorithm, LEarning to Plan before Answering (LEPA), that learns to generate anticipatory plans before generating detailed step-by-step problem solutions. The anticipatory plans serve as high-level abstract meta-knowledge that outlines the solution generation path and shields the LLM from the distraction of irrelevant details. During data generation, LEPA prompts the LLM to first devise an anticipatory plan that encapsulates the high-level problem-solving steps, and then generate a solution that aligns with both the problem and the plan. If the solution is correct, the plan-solution pair is stored into the training dataset. Otherwise, the LLM is asked to reflect on the plan and the incorrect solution, and refine the plan until it successfully prompts the LLM to generate correct solutions. With this self-reflection mechanism, LEPA acquires plans that are instrumental in yielding correct solutions. During model optimization, we utilize supervised fine-tuning (SFT) to train the LLM to predict both the plans after self-reflection and the corresponding solutions. As shown in Figure 1, after self-training with LEPA, the LLM generates helpful abstract anticipatory plans that outline the solution steps and are generalizable to similar problems, thus achieving better performance than baseline algorithms. LEPA is extensively evaluated on various challenging language reasoning benchmarks including Hendrycks MATH, and significantly outperforms baseline methods.

To summarize, our main contributions are listed as follows:

1. We present the fundamental problem of what information should self-generated data encapsulate in the field of LLM self-training.
2. We propose a novel self-training algorithm, LEPA, that learns to generate anticipatory plans, which serves as high-level abstract meta-knowledge guiding solution generation, before generating detailed problem solutions.

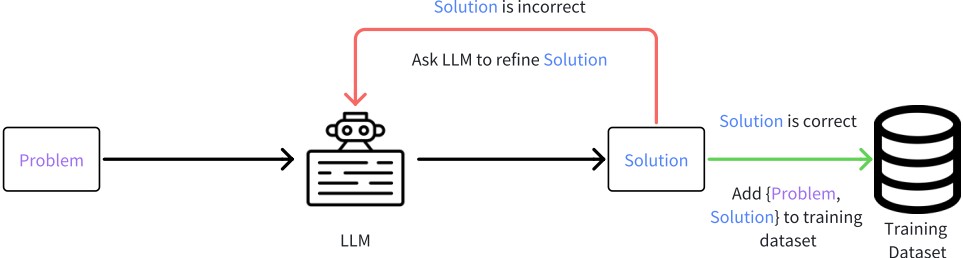

(a) Baseline algorithms' data generation procedure.

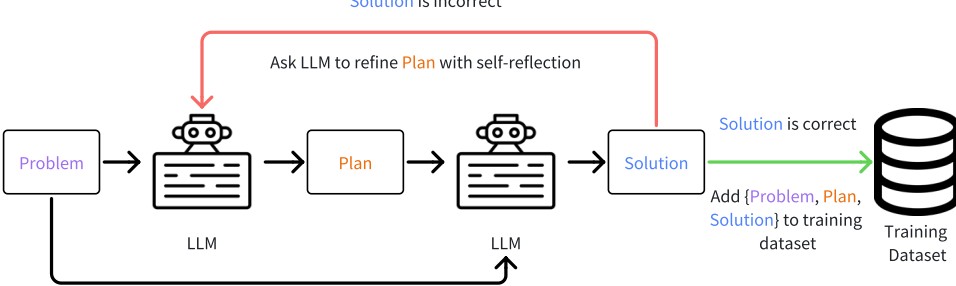

(b) LEPA's data generation procedure.

Figure 2: Comparison between baseline algorithms' and LEPA's data generation procedure. (a) Baseline algorithms only generate step-by-step solutions to each problem, lacking high-level abstract meta-knowledge that guides solution generation. (b) LEPA generates anticipatory plans before generating detailed problem solutions. These plans are optimized with self-reflection, and encapsulate the high-level abstract problem-solving steps. The plans efficiently guide the LLM to generate correct solutions.

3. We evaluate LEPA on several challenging language reasoning benchmarks and demonstrate LEPA's superior performance compared to based algorithms.

## 2 LEARNING TO PLAN BEFORE ANSWERING (LEPA)

This section introduces LEPA, a novel self-training algorithm that self-trains the LLM to devise high-level anticipatory plans, which serve as abstract solution-generation blueprints, before generating detailed problem solutions. LEPA iterates between a data generation phase and a model optimization phase. In the data generation phase, LEPA generates high-quality plan-solution pairs with self-reflection. In the model optimization phase, LEPA fine-tunes the LLM with the generated data using SFT. Finally, we discuss multiple advantages that the anticipatory plans offer for enhancing the self-training process.

### 2.1 DATA GENERATION PHASE

LEPA operates within the common self-training framework, which involves an initial LLM denoted as $\theta_0$, a set of prompts containing $N$ problems $\mathcal{D}_{prompt} = \{x_i\}_{i=0}^{N-1}$, and a binary scoring function $f_{cor}(x_i, y_i)$ that evaluates the correctness of a solution $y_i$ with a score of either 0 or 1.

In each iteration $t$, as depicted in Figure 2, LEPA differs from previous methods in that it does not directly prompt the LLM to generate step-by-step solutions to problems. Instead, LEPA instructs the LLM to first generate an anticipatory plan $p_i^t$ that serves as an abstract blueprint for solution generation, and then generate the actual solutions $y_i^t$ based on the plan and the problem. To avoid the degenerate case of generating plans containing detailed step-by-step problem solutions, LEPA

stresses in the prompt that the plan should be general high-level meta-knowledge that is applicable to similar problems, and should not contain any problem-specific information such as detailed calculations. If the solution is correct, i.e., $r_{cor}(x_i, y_i) = 1$, then the problem-plan-solution tuple $(x_i, p_i^t, y_i^t)$ is added to the training dataset $\mathcal{D}_{train}^t$. Otherwise, LEPA refines the plan with self-reflection. The LLM is prompted with the problem, the previous plan, the corresponding incorrect solution, and the correct answer (if accessible). Then LEPA instructs the LLM to reflect on why the previous plan fails to guide itself to generate correct solutions, and then generate a new plan based on its reflection results. To avoid information bypassing, LEPA also stresses in the reflection prompt that the reflected plan should not contain problem-specific information, including detailed calculation and the correct answer. LEPA evaluates the refined plan by instructing the LLM to solve the problem with the refined plan. If the generated solution is correct, the problem-plan-solution tuple $(x_i, p_i^t, y_i^t)$ is added to the training dataset. Otherwise, LEPA repeats the self-reflection process, unless either a correct solution is generated or the number of trials reaches a certain limit $l$. The self-reflection process empowers LLMs to enhance anticipatory plans based on correctness feedback and analysis of unsuccessful attempts, thus efficiently seeking out superior plans.

## 2.2 MODEL OPTIMIZATION PHASE

In each iteration, after acquiring the training dataset $\mathcal{D}_{train}^t$, LPEA optimizes the model with SFT. LEPA formats data into a two-round conversation. In the first round, The user inputs the problem $x_i$ and requires the LLM to generate an anticipatory plan, and the assistant output is the plan $p_i^t$. In the second round, the user instructs the LLM to solve the problem based on the plan it proposed, and the assistant output is the solution $y_i^t$. The training objective is to minimize the following negative log-likelihood loss:

$$\mathcal{L}_{SFT}(\theta_t, \mathcal{D}_{train}^t) = -\mathbb{E}_{(x_i, p_i^t, y_i^t) \sim \mathcal{D}_{train}^t}[\log p_{\theta_t}(p_i^t, y_i^t | x_i)]. \tag{1}$$

While we employ SFT for algorithm simplicity, LEPA is also compatible with more sophisticated reinforcement learning (RL) algorithms such as Direct Policy Optimization (DPO) (Rafailov et al., 2024) and Proximal Policy Optimization (PPO) (Schulman et al., 2017). We believe RL algorithms can further boost LEPA's performance, and are important future directions. The pseudo-code for LEPA is presented in Algorithm 1. Detailed prompts and hyper-parameters used by LEPA is deferred to Appendix A.

## 2.3 WHY IS THE ANTICIPATORY PLAN BENEFICIAL?

Central to LEPA's efficacy is the anticipatory plan, offering multiple advantages for self-training. This subsection discusses these benefits in detail.

**Reducing cognitive workload.** As demonstrated in Figure 1, without the anticipatory plans, the LLM may get lost in the problem-solving process, leading to erroneous solution steps. In contrast, the anticipatory plans serve as blueprints that outline the necessary problem-solving steps, and shield the LLM from the distraction of irrelevant details. Consequently, when generating detailed problem solutions, the LLM is conscious of what to do at the current step, and successfully solves the problem. Research in cognitive science (Wang & Chiew, 2010; Radüntz, 2020) supports the notion that such a structured approach significantly eases cognitive load and improves learning efficiency.

**Learning generalizable high-level meta-knowledge.** The anticipatory plans are abstract high-level meta-knowledge that does not involve problem specifics, and is thus generalizable across similar problems. For example, the plan demonstrated in Figure 1 can be readily adapted to a variety of combinatorial mathematical problems with similar underlying structures but different parameters. From the meta-learning perspective, LEPA can be interpreted as a meta-learning algorithm that extracts the meta-knowledge in the form of anticipatory plans. The learned meta-knowledge empowers the LLM to solve similar problems more effectively.

**Learning generalizable high-level meta-knowledge.** When the correct answer is accessible, the anticipatory plans enable self-reflection that avoids the pitfall of information bypassing. Previous methods like STaR (Zelikman et al., 2022) directly modify incorrect solutions by referring to the

---

**Algorithm 1** LEPA: LEarning to Plan before Answering

---

1: **Require:** An initial LLM $\theta_0$, a set of problems $\mathcal{D}_{prompt} = \{x_i\}_{i=0}^{N-1}$, a binary scoring function $f_{cor}(x_i, y_i)$, number of iterations $T$, maximum self-reflection trials $l$, learning rate $\alpha$
2: **for** $t \leftarrow 0$ **to** $T-1$ **do**                                 // In each iteration do
3:     Initialize an empty training set $\mathcal{D}_{train}^t$
4:     **for** $i \leftarrow 0$ **to** $N-1$ **do**                             // For each problem do
5:         Ask $\theta_t$ to generate anticipatory plan $p_i^{t,0}$ to problem $x_i$
6:         Ask $\theta_t$ to generate solution $y_i^{t,0}$ based on $x_i$ and $p_i^{t,0}$
7:         **if** $f_{cor}(x_i, y_i^{t,0}) == 1$ **then**             // Solution is correct, add to training set
8:             Add $\{x_i, p_i^{t,0}, y_i^{t,0}\}$ to $\mathcal{D}_{train}^t$
9:         **else**
10:             **for** $j \leftarrow 1$ **to** $l$ **do**                   // Self-reflection iterations
11:                 Ask $\theta_t$ to self-reflect on $p_i^{t,j-1}$ and $y_i^{t,j-1}$, and generate $p_i^{t,j}$
12:                 Ask $\theta_t$ to generate solution $y_i^{t,j}$ based on $x_i$ and $p_i^{t,j}$
13:                 **if** $f_{cor}(x_i, y_i^{t,j}) == 1$ **then**
14:                     Add $\{x_i, p_i^{t,j}, y_i^{t,j}\}$ to $\mathcal{D}_{train}^t$
15:                     **Break**             // Solution is correct, stop self-reflection
16:                 **end if**
17:             **end for**
18:         **end if**
19:     **end for**
20:     $\theta_{t+1} \leftarrow \theta_t - \alpha \nabla_{\theta_t} \mathcal{L}_{SFT}(\theta_t, \mathcal{D}_{train}^t)$         // Model Optimization with SFT
21: **end for**

---

correct answer, and are very likely to cheat by only modifying the final answer and ignoring the consistency between intermediate steps and the final answer (Singh et al., 2023). In contrast, as LEPA requires the anticipatory plans to not include any problem-specific information including the final correct answer, it isolates the correct answer from solution generation. The model must generate correct solutions without seeing the correct answer, preventing the model from cheating during solution generation.

## 3 EXPERIMENTS

To demonstrate the effectiveness of LEPA, we evaluate on several challenging reasoning benchmarks, including Hendrycks MATH (challenging math problems) (Hendrycks et al., 2021), Hellaswag (sentence completion reasoning) (Zellers et al., 2019), BoolQ (paragraph understanding and reasoning) (Clark et al., 2019), and PIQA (physics reasoning) (Bisk et al., 2020). For Hendrycks MATH, we evaluate solution correctness with the function provided by the dataset creators (`https://github.com/hendrycks/math`).We utilize Llama 3 8B Instruct (Dubey et al., 2024) as the initial LLM. LEPA is compared against several representative self-training algorithms: $ReST$ (Gulcehre et al., 2023), $ReST^{EM}$ (Singh et al., 2023), and STaR (Zelikman et al., 2022). All these baseline methods only generate step-by-step solutions to problems. Both $ReST$ and $ReST^{EM}$ generate solutions with rejection sampling. In each iteration, $ReST$ fine-tunes the model trained after the previous iteration, while $ReST^{EM}$ instead fine-tunes from the initial LLM. STaR generates solutions by prompting the LLM to modify incorrect solutions with the aid of correct answers, and also fine-tunes from the initial LLM in each iteration. We demonstrate algorithms' test accuracy at convergence[1]. For a fair comparison, all methods do not utilize few-shot examples in their prompts. We also demonstrate the initial LLM's efficacy, with either a zero-shot CoT prompt (Kojima et al., 2022) or a LEPA prompt that instructs it to first generate an anticipatory plan before answering.

---

[1]As STaR's test accuracy drops significantly on MATH, we instead demonstrate its highest test accuracy.

Table 1: Test accuracy of LEPA and various baselines on four challenging reasoning benchmarks. "CoT" and "Plan+CoT" refer to the initial LLM's performance with a zero-shot CoT prompt and the LEPA prompt, respectively. LEPA demonstrates superior accuracy in comparison to all other algorithms on each of the benchmarks. Numbers in the parentheses are LEPA's performance improvement over the best-performing baseline algorithm on each benchmark.

| | CoT | Plan+CoT | $ReST$ | $ReST^{EM}$ | STaR | LEPA |
|---|---|---|---|---|---|---|
| Hellaswag | 60.8% | 56.1% | 86.3% | 86.4% | 85.7% | **91.2% (+4.8%)** |
| Hendrycks MATH | 19.5% | 22.1% | 28.2% | 27.2% | 25.9% | **30.2% (+2.0%)** |
| BoolQ | 77.3% | 80.8% | 84.5% | 86.3% | 85.8% | **88.4% (+2.1%)** |
| PIQA | 67.0% | 75.7% | 81.4% | 83.5% | 84.2% | **85.9% (+1.7%)** |
| Average | 56.1% | 58.7% | 70.1% | 70.8% | 70.4% | **73.9% (+3.1%)** |

## 3.1 MAIN RESULTS

Table 1 presents a comparative analysis of algorithm performance across the four reasoning benchmarks. Notably, in the absence of self-training, the LEPA prompt (Plan+CoT) enhances the initial LLM's performance on three benchmarks when compared to the traditional zero-shot CoT prompt (CoT). This suggests that the practice of formulating anticipatory plans before generating detailed solutions can significantly improve model efficacy. However, on the Hellaswag benchmark, Plan+CoT falls short of CoT, implying that such enhancement is not uniformly achievable across different tasks, potentially due to the initial LLM's lack of calibration for producing high-quality anticipatory plans. As for self-training performance, baseline self-training algorithms only train the LLM to predict step-by-step solutions, lacking abstract high-level meta-knowledge about problem-solving. As a consequence, these algorithms perform poorly on these benchmarks. In contrast, LEPA efficiently extracts high-level abstract meta-knowledge with the anticipatory plans, thereby surpassing all baseline algorithms consistently across all benchmarks.

Figure 3 illustrates algorithms' learning curve across learning iterations. LEPA's superior performance is evident across all benchmarks. Specifically, on Hellaswag, LEPA lags initially during the early iterations (0-10), where the LEPA prompt is slightly less effective than the zero-shot CoT prompt. However, as training progresses, LEPA's performance incrementally surpasses that of the baseline algorithms, suggesting that self-training is instrumental in awakening the LLM's capacity to conceive and leverage anticipatory plans effectively. On the remaining three benchmarks, LEPA acquires better initial performance and converges at higher test accuracies, demonstrating the effectiveness of introducing the anticipatory plans. We also observe a great performance drop of STaR on Hendrycks MATH. This is because STaR is very likely to generate false-positive solutions, i.e., solutions with wrong rationales but correct final answers (Singh et al., 2023), and greatly hinders learning on complex reasoning benchmarks like Hendrycks MATH.

## 3.2 ABLATION STUDIES

LEPA consists of three key components: the anticipatory plan, plan optimization with self-reflection, and utilizing more inference compute to achieve better performance. This subsection discusses the necessity of each component with ablation studies.

**Anticipatory plans.** We test a variant of LEPA that does not introduce anticipatory plans in the data generation phase, and only trains the LLM to predict the step-by-step solutions optimized with self-reflection. As shown in Table 2, this variant ("Without Plan") under-performs LEPA. There are two reasons for this degrade in performance. Firstly, without the anticipatory plans, the LLM does not learn abstract high-level meta-knowledge about problem-solving. Secondly, as discussed in Section 2.3, directly performing self-reflection on the solutions is very likely to generate false-positive solutions, which greatly hiders learning.

**Self-reflection.** To demonstrate the necessity of self-reflection in LEPA's plan optimization, we test a variant that instead utilizes rejection sampling (Singh et al., 2023) to sample plan-answer pairs. As shown in Table 2, this variant ("Without Self-Reflection") also performs worse than LEPA.

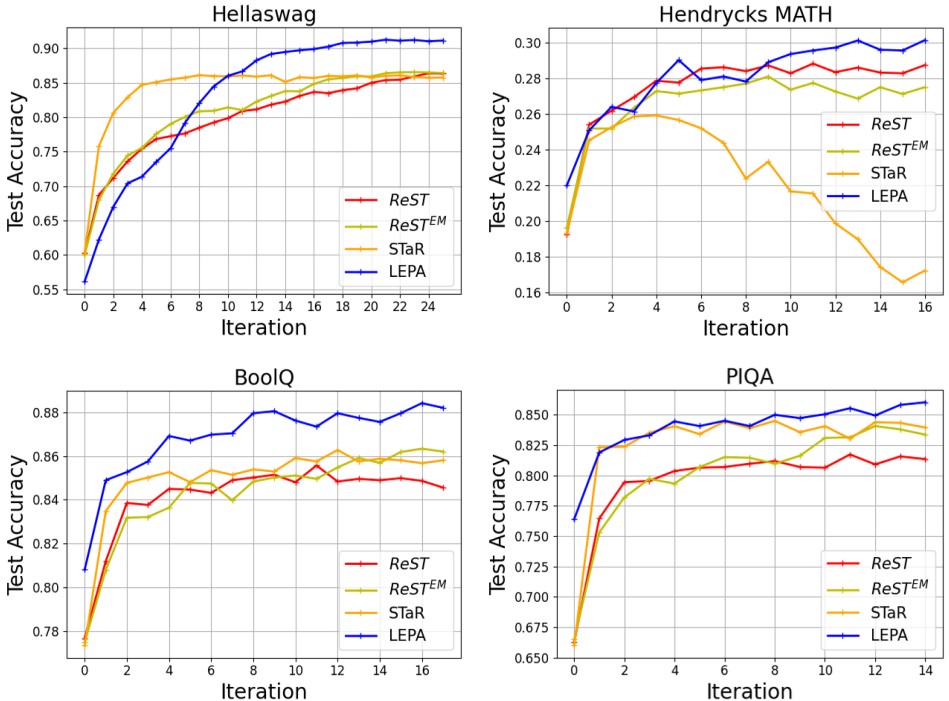

Figure 3: Algorithms' learning curves on the four benchmarks. LEPA achieves better performance than baseline algorithms.

Table 2: Ablation study on the anticipatory plan and self-reflection. We also demonstrate the performance of $ReST^{EM}$, the baseline with the highest average test accuracy. "Without Plan" is LEPA without anticipatory plans, and "Without Self-Reflection" is LEPA without self-reflection.

|                  | $ReST^{EM}$ | LEPA     | Without Plan | Without Self-Reflection |
| ---------------- | ----------- | -------- | ------------ | ----------------------- |
| Hendrycks MATH   | 27.2%       | **30.2%** | 24.3%        | 28.8%                   |
| BoolQ            | 86.3%       | **88.4%** | 84.8%        | 86.9%                   |
| PIQA             | 84.2%       | **85.9%** | 84.5%        | 84.8%                   |

This result implies that self-reflection is more effective than rejection sampling in optimizing the anticipatory plans, as it gives linguistic feedback for LLMs to improve the previous plans.

**Different ways of utilizing inference compute.** LEPA generates both anticipatory plans and problem solutions, utilizing more compute at inference time. it is worth discussing how much contribution the extra compute makes, and whether the anticipatory plan is an effective way to utilize inference compute. For the first question, as discussed in Section 3.2, without self-training, utilizing inference compute with anticipatory plans can improve performance on three of the four benchmarks, and degrade performance on one benchmark. In contrast, after self-training, the anticipatory plans can consistently help LEPA outperform baseline methods. This result demonstrates that extra inference compute contributes a part to LEPA's performance, and self-training is also vital for unlocking the LLM's ability to efficiently utilize these extra compute. For the second question, we test three other variants that train the LLM to utilize inference compute in different ways. The first variant adds silence tokens in the solution to give the LLM more compute to generate answers (Goyal et al., 2023). The second variant trains the LLM to first output a solution, and then outputs a new solution if it finds the original solution incorrect. For data generation of this variant, solutions are generated with rejection sampling, analogous to $ReST$. We synthesize training data by appending correct solutions to the end of incorrect solutions. The third variant simply asks the LLM to generate long solutions. All variants fine-tune the LLM with $ReST$. As shown in Table 3, LEPA is the only method that successfully utilizes additional inference compute to outperform baseline methods. In

Table 3: Ablation study on ways of utilizing inference compute. We test on the Hendrycks MATH dataset."Silence token" is the variant that adds silence tokens in the solution. "Correction" is the variant that trains the LLM to output new solutions if it finds its initial solution incorrect. "Long Solution" is the variant that instructs the LLM to generate long solutions. "# of Tokens" is the average token length of the LLM's responses to test problems, and "Accuracy" is the LLM's test accuracy. LEPA is the only method that efficiently utilizes additional inference compute to outperform baseline methods. We put the results in two rows due to the page width limit.

| STaR | | *ReST* | | LEPA | |
|---|---|---|---|---|---|
| # of Tokens | Accuracy | # of Tokens | Accuracy | # of Tokens | Accuracy |
| 175.1 | 25.9% | 477.8 | 28.2% | 826.4 | **30.2%** |
| Silence Tokens | | Correction | | Long Solution | |
| # of Tokens | Accuracy | # of Tokens | Accuracy | # of Tokens | Accuracy |
| 869.3 | 28.3% | 979.4 | 27.8% | 1409.7 | 25.4% |

contrast, the first variant performs similarly to the *ReST* baseline, suggesting that silence tokens offer limited benefits for the challenging Hendrycks MATH benchmark. Both the second and the third variant underperform *ReST*, as the LLM is trained to predict tokens with scant correlation to correct solution formulation. The results above implies that introducing the anticipatory plans is a more efficient way to generate long self-training data compared to the aforementioned alternatives. Detailed implementation of these variants are deferred to Appendix B.

**Incorporation with RL algorithms.** To demonstrate that LEPA is also applicable to more advanced RL optimization algorithms, we test a variant of LEPA that utilizes REINFORCE (Zhang et al., 2021b) as the underlying optimization algorithm, which is called LEPA+REINFORCE. The only difference between LEPA and LEPA+REINFORCE is that LEPA+REINFORCE labels data with rewards of either 1 or -1 (based on the final answer correctness), and optimizes the LLM with the labelled data using the REINFORCE algorithm. On Hendrycks MATH, LEPA+REINFORCE achieves a test accuracy of 30.6%, while the original LEPA achieves 30.2%. This performance improvement demonstrates the potential of incorporating LEPA with more advanced optimization algorithms like RL, and is a promising future direction.

**LPEA with RL.** To investigate the potential of incorporating LPEA with RL algorithms, we demonstrate a variant of LEPA that utilizes REINFORCE as the optimization algorithm in the model optimization phase. This variant is named LEPA+REINFORCE. The only difference from LEPA is that LEPA+REINFORCE does not discard failure data. Instead, it utilizes the solution correctness as the reward (1 for correct solutions, -1 for incorrect solutions). This implementation makes no modification to the data generation process. On Hendrycks MATH, LEPA+REINFORCE achieves a test accuracy of 30.6%, while LEPA achieves 30.2%. This performance improvement demonstrates the feasibility and effectiveness of incorporating LEPA with RL algorithms. Additional ablation studies including algorithm performance on OOD benchmarks, other LLMs, additional benchmarks, and evaluation with Simple-Eval are deferred to Appendix C.

### 3.3 CASE STUDY

We present a case to demonstrate how LEPA's self-reflection mechanism optimizes the anticipatory plans and the solutions. As shown in Figure 4, the initial plan generated by the model is too vague, and cannot provide enough guidance for the model to solve the problem correctly. Consequently, during solution generation, the model generates irrelevant steps, makes a mistake in the symbol of the expression, and fails to answer correctly. In the self-reflection process, the model finds out that the previous answer failed to calculate the correct symbols. So it modifies the plan to contain more detailed instructions on how to solve this problem. Note that the plan after self-reflection is still general meta-knowledge that is applicable to a wide range of similar problems. With this modified plan, the model pays more attention to signs, generates only necessary steps, and successfully generates a correct solution.

**Question:**
What is the value of $\sqrt{15 - 6\sqrt{6}} + \sqrt{15 + 6\sqrt{6}}$?

**Initial Plan:**
Analyze the problem carefully. Identify the mathematical object or expression involved, and determine the relevant mathematical field or discipline it belongs to. In this case, identify the type of expression, its structure, and the mathematical operations involved. Consider the properties and relations between the given numbers, expressions, and variables. Look for any opportunities to apply mathematical concepts, theorems, or formulas to simplify or solve the problem.

**Answer guided by the initial plan:**
Analyzing the problem carefully, I identify the mathematical object as a radical expression involving square roots. The relevant mathematical field is algebra, specifically rationalizing the denominators and simplifying expressions.
Step 1: I consider the structure of the expression and identify the perfect squares within the radicals. Recognizing that $6\sqrt{6}$ can be rewritten as $\sqrt{36}\sqrt{6} = 6\sqrt{6}$, I can rewrite the expression as $\sqrt{15 - 6\sqrt{6}} + \sqrt{15 + 6\sqrt{6}} = \sqrt{9 + 6 - 6\sqrt{6}} + \sqrt{9 + 6 + 6\sqrt{6}}$.

I apply the difference of squares formula to the first radical, obtaining $\sqrt{15 - 6\sqrt{6}} = \sqrt{\left(3 + \sqrt{6}\right)^2} = 3 + \sqrt{6}$. Similarly, I apply the difference of squares

formula to the second radical, obtaining $\sqrt{15 + 6\sqrt{6}} = \sqrt{\left(3 + \sqrt{6}\right)^2} = 3 + \sqrt{6}$.
Step 3:
I combine the expressions, adding the like terms, to obtain $3 + \sqrt{6} + 3 + \sqrt{6} = 6 + 2\sqrt{6}$.
Answer: $6 + 2\sqrt{6}$ ❌

**Self-reflection:**
The model fails to solve the problem because it incorrectly applies the difference of squares formula, resulting in the same expression for both radicals. This is likely due to the model's lack of attention to the signs and the correct application of the formula.
The new instruction should focus on the correct application of the difference of squares formula and the importance of considering the signs.

**Plan after self- reflection:**
When simplifying radical expressions, consider the structure of the radicand and the signs.
1. Identify the radical expression and its components.
2. Simplify each radical separately, considering the signs.
3. Combine like terms to obtain the simplified expression.
4. Verify the result by checking the original expression and the simplified expression for equivalence.

**Answer guided by the plan after self-reflection:**
Step 1: Identify the radical expression and its components.
The radical expression is $\sqrt{15 - 6\sqrt{6}} + \sqrt{15 + 6\sqrt{6}}$, which consists of two radical terms.
Step 2: Simplify each radical separately, considering the signs.

We can start by simplifying the first radical: $\sqrt{15 - 6\sqrt{6}} = \sqrt{\left(3 - \sqrt{6}\right)^2} = 3 - \sqrt{6}$.

Similarly, we can simplify the second radical: $\sqrt{15 + 6\sqrt{6}} = \sqrt{\left(3 + \sqrt{6}\right)^2} = 3 + \sqrt{6}$.
Step 3: Combine like terms to obtain the simplified expression.
The simplified expression is $3 - \sqrt{6} + 3 + \sqrt{6} = 6$.
Step 4: Verify the result by checking the original expression and the simplified expression for equivalence.
The original expression and the simplified expression are equivalent, so our answer is correct.
Answer: 6 ✅

Figure 4: A case study demonstrating how LEPA optimizes the anticipatory plans and the solutions with self-reflection. The initial plan is too broad and lacks detail, and fails to provide enough guidance to generate correct answers. The self-reflection process successfully analyses what is wrong, and generates a new, high-quality plan that provides more guidance while maintaining generalizability. With the new plan after self-reflection, the model successfully generates correct solutions.

# 4 RELATED WORKS

**Self-training.** With the fast development of LLMs, the thirst for data continues to grow. A promising way is to generate high-quality data with the LLM itself. A branch of works mainly focus on designing the data generation progress. STaR (Zelikman et al., 2022) operates by initially prompting the LLM to produce step-by-step solutions, followed by an adjustment phase where the LLM corrects its errors with the aid of the correct answers. One severe limitation of STaR is that the modification process makes it very possible to generate false-positive solutions, i.e., solutions with wrong rationales but correct final answers. RFT (Yuan et al., 2023), $ReST$ (Gulcehre et al., 2023), and $ReST^{EM}$ (Singh et al., 2023) instead adopt rejection sampling for data generation, and suffer less from the false-positive issue. TRICE (Hoffman et al., 2024) improves over STaR by utilizing a Markov-chain Monte Carlo expectation-maximization algorithm to sample solutions, and introducing a control-variate method to control gradient variance. Re-ReST (Dou et al., 2024) utilizes self-reflection to correct the generated wrong answers. LMSI (Huang et al., 2022) considers the scenario where the correctness of model-generated data cannot be verified during training, and filers data with majority voting. Apart from these methods, SPAG (Cheng et al., 2024) generates data by asking LLMs to self-play in adversarial games. These previous methods above only generate step-by-step solutions to problems, and lack high-level meta-knowledge that are generalizable across

similar problems. In contrast, LEPA learns abstract meta-knowledge in the form of anticipatory plans, and achieves better performance on complex benchmarks.

**Scaling inference compute.** As proposed by Snell et al. (2024) and confirmed by the recent inspiring GPT O1 model (Hu et al., 2024), scaling inference compute can further boost LLM performance. Similar to LEPA, PS Prompting (Wang et al., 2023b) also scales inference compute by asking the LLM to first generate a plan before answering, but does not consider how to generate data and fine-tune the LLM. Moreover, it does not consider how to automatically optimize the anticipatory plans. HSP (Fu et al., 2024) is the most relevant work to ours, which trains the LLM to output hints before solving the problem. However, HSP's hints are pre-collected rather than self-generated, and induce additional data collection costs. PHP (Zheng et al., 2023) utilizes previously generated answers as hints, and encourages the LLM to answer with reference to its previous answers. LEPA efficiently utilizes inference compute by training the LLM to generate helpful anticipatory plans, which contain high-level meta-knowledge on problem-solving, before generating actual problem solutions. These plans are automatically optimized by the LLM itself, and do not require additional human design.

**Meta-learning.** Meta-learning aims at "learning to learn", i.e., designing meta-algorithms that optimize learning algorithms automatically (Finn et al., 2017; Sung et al., 2017; Rakelly et al., 2019; Zhang et al., 2021a; Wang et al., 2023a). LEPA can be interpreted as a meta-learning algorithm that learns the meta-knowledge of designing the anticipatory plans for each problem, rather than designing plans with human effort. The most relevant work is Quiet-STaR (Zelikman et al., 2024), which meta-learns meta-tokens that help the LLM to predict the next token. LEPA considers the setting of problem-solving rather than general next-token prediction, and meta-learns the generation of anticipatory problem-solving plans.

**Planning in LLMs.** Recently, several works have demonstrated the effectiveness of integrating planning in LLMs. ReAct (Yao et al., 2022) and DEPS (Wang et al., 2024c) generate plans before dealing with decision-making problems, and LUMOS (Yin et al., 2023) fine-tunes the LLM on pre-collected datasets containing planning data. To our best knowledge, LEPA is the first work to integrate planning in the process of self-training, and improves the LLM's planning ability by training on self-generated data.

**Self-reflection.** Self-reflection enables LLMs to reflect on their mistakes and generate better responses. It can be viewed as a process of in-context optimization to produce better responses. Previous works demonstrate that self-reflection can significantly improve LLM response quality (Renze & Guven, 2024; Shinn et al., 2024; Madaan et al., 2024). LEPA utilizes self-reflection to optimize plans and solutions in the data generation phase, and acquires data of higher quality.

## 5 CONCLUSION

This paper presents the fundamental problem of what data should be generated in self-training algorithms. Inspired by cognitive science research and recent meta-learning advances, we propose a novel idea of learning abstract meta-knowledge in the form of anticipatory problem-solving plans. Based on this idea, we propose a novel self-training algorithm, LEPA, which automatically generates and learns the anticipatory plans. Experiment results on several challenging reasoning benchmarks demonstrate the effectiveness of LEPA. An interesting future direction is to incorporate LEPA with more advanced model optimization methods such as RL. It is also worth exploring how well can LEPA perform on larger and more advanced LLMs, and how to scale LEPA to utilize more inference compute. Furthermore, as LLMs may solve simple problems without planning, an important future direction is to automatically identify complex problems that require planning from simple problems that can be easily solved without planning. This identification can avoid wasting compute resources and help the LLM solve problems more efficiently.

## ACKNOWLEDGEMENT

This work is supported by National Natural Science Foundation of China (62176135), the National Key R&D Program of China (2022ZD0161700), Shanghai Qi Zhi Institute Innovation Program SQZ202306 and the Tsinghua University Dushi Program.

## ETHICS STATEMENT

Concerns about safety and reliability are key points of discussion in the LLM community. The use of anticipatory plans in LLMs is a step towards making the models' actions more understandable and transparent to people. Yet, LEPA cannot guarantee that every solution will strictly match the plans it creates, which means further work is needed to solidify the trustworthiness of LLMs.

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

## A    DETAILED PROMPTS AND HYPER-PARAMETERS

This section demonstrates the detailed prompts and the hyper-parameters used by LEPA and baseline algorithms. Figure 5 presents the prompts used by LEPA and baseline algorithms.

As for hyper-parameters, for a fair comparison, we ensure that all algorithms have the same number of trials (5) in the data generation phase. LEPA is allowed to have maximally 4 self-reflection processes for each problem. For $ReST$ and $ReST^{EM}$, 5 solutions are sampled for each question. For STaR, it has maximally 4 opportunities to modify the previous incorrect answer. All algorithms fine-tunes the LLM for one epoch in each model optimization phase. For the data generation phase of all algorithms, we use a temperature of 0.5 for sampling. We use a temperature of 0.0005 for all test results. We use 3e-7 as the learning rate for all learning algorithms.

**Prompt for anticipatory plan generation:**
You are an expert at designing plans for large language models to solve problems. The problem to be solved is:
[Question]
Output the plan you design. Note that the plan should be general knowledge that help solve similar problems, so do not contain any question-specific information. Also, the content will be directly added to the prompt, so pay attention to its format. The plan should be concise, no longer than 1024 tokens. Output only the plan. Do not output any other words.

**Prompt for solution generation:**
Based on the plan you propose, solve the problem step by step. In each step of your solution, explain how the plan affect you to form your answers. The last line of your response should be of the form Answer: $ANSWER (without quotes) where $ANSWER is the answer to the problem. Remember to put your answer on its own line after "Answer:", and you do not need to use a \\boxed command. Your response should be concise, no longer than 1024 tokens. The problem is:
[Question]

**Prompt for self-reflection:**
You are an expert in designing plans for large language models to solve problems. You have found that the original plan fails to solve a problem. You need to analyze the failure case, and design a new plan. The new plan should help the large language model to solve the failure case. You are encouraged to design plans distinct from the original plan to better explore high-quality plans.
The problem is:
[Question]
The original plan is:
[Original Plan]
The incorrect solution  given by the large language model under the original plan is:
[Original solution]
The desired correct final answer is:
[Correct Answer]
Analyze the information above. Why does the model fail to solve the problem? What is wrong in the answer? How to design a new plan so that the model can correctly solve the problem? How distinct should the new plan be from the original plan? What contents should the new plan obtain? Pay special attention to the formatting requirements. Does the model's output strictly follow the required output format? Answer concisely, no longer than 2560 tokens.

**Prompt for new plan generation after self-reflection:**
Based on the analysis above, output the new plan. Note that the new plan should be general knowledge that help solve similar problems, so do not contain any task-specific information. You must not contain the correct final answer in the plan. You are encouraged to design plans distinct from the original plan to better explore high-quality plans. Also, the content will be directly added to prompt, so pay attention to its format. The content should be short and concise, no longer than 1024 tokens. Output only the plan. Do not output any other words.

(a) LEPA prompt.

**Prompt for solution generation:**
Solve the following problem step by step. The last line of your response should be of the form Answer: $ANSWER (without quotes) where $ANSWER is the answer to the problem. Remember to put your answer on its own line after "Answer:", and you do not need to use a \\boxed command. Your response should be concise, no longer than 1024 tokens. The problem is:
[Question]

**Prompt for solution modification (only used in STaR):**
Your solution is wrong. The correct answer is:
[Correct Answer]
Modify your previous solution to get the correct answer. Output the modified solution only. Do not output any other words.

(b) Prompt used by baseline methods.

Figure 5: Detailed prompts used by (a) LEPA and (b) baseline algorithms.

## B    ABLATION DETAILS

This section presents the details of the variants discussed in the "Different ways of utilizing inference compute" part of Section 3.2.

For the second variant, we first sample correct and incorrect solutions for each problem with rejection sampling. Then we synthesize training data by first adding a sentence of "Oops, I made a mistake. The correct solution is: " to the end of incorrect solutions. Then we append a correct solution to the end of this sentence.

For the third variant, we explicitly instruct the LLM to output solutions that are approximately 2,000 words long. We observe that the LLM generates verbose responses that obscure the important steps in solving the problem.

## C    ADDITIONAL ABLATION STUDIES

**OOD performance.**    We evaluate OOD performance by training on Hendrycks MATH and testing on the Math category of MMLU-Pro (Wang et al., 2024a). As shown in Table 4, LEPA consistently outperforms baseline algorithms in this OOD setting.

|  | CoT | Plan+CoT | $ReST$ | $ReST^{EM}$ | STaR | LEPA |
|---|---|---|---|---|---|---|
| Performance | 30.4% | 33.9% | 35.1% | 35.3% | 35.8% | **38.9%** |

Table 4: Performance of different algorithms training on Hendrycks MATH and testing on the Math category of MMLU-Pro. "CoT" and "Plan+CoT" refer to the initial LLM's performance with a zero-shot CoT prompt and the LEPA prompt, respectively. LEPA achieves better generalization than baseline algorithms.

**Other LLMs.**    We additionally evaluate algorithm performance on Llama 3.1 8B Instruct. As shown in Table 5, on the Hendrycks MATH dataset, the LEPA prompt can slightly improve over the zero-shot CoT prompt on the initial LLM. As for self-training, LEPA significantly outperforms the baseline algorithm. These empirical results are consistent with our main results presented in Section 3.1.

**Additional Benchmarks.**    We additionally evaluate on CSQA (Saha et al., 2018) and MMLU (Hendrycks et al., 2020), and results are shown in Table 6. LEPA consistently outperforms baseline algorithms on these benchmarks.

| Algorithm | CoT | Plan+CoT | $ReST$ | $ReST^{EM}$ | STaR | LEPA |
|---|---|---|---|---|---|---|
| Performance | 37.2% | 38.4% | 45.3% | 46.9% | 45.0% | **49.6%** |

Table 5: Algorithm performance on Hendrycks MATH, with Llama 3.1 8B Instruct as the initial LLM. "CoT" and "Plan+CoT" refer to the initial LLM's performance with a zero-shot CoT prompt and the LEPA prompt, respectively

|  | CoT | Plan+CoT | ReST | $ReST^{EM}$ | STaR | LEPA |
|---|---|---|---|---|---|---|
| CSQA | 67.1% | 69.3% | 73.2% | 74.0% | 74.1% | **75.2%** |
| MMLU | 61.9% | 60.1% | 64.3% | 65.6% | 65.8% | **66.1%** |

Table 6: Performance comparison of different methods on CSQA and MMLU benchmarks. LEPA achieves higher performance than baseline algorithms.

**Evaluation with Simple-Eval.**    We re-evaluate Hendrycks MATH performance with Simple-Eval, and the results are demonstrated in Table 7. With the new evaluation, LEPA still outperforms baseline algorithms.

| CoT | Plan+CoT | $ReST$ | $ReST^{EM}$ | STaR | LEPA |
|------|----------|--------|-------------|-------|--------|
| 26.1% | 28.5% | 31.2% | 31.4% | 29.2% | **33.7%** |

Table 7: Hendrycks MATH performance evaluated with Simple-Eval. With the new evaluation, LEPA still outperforms baseline algorithms.

