# OpenReview forum: "Learning to Plan Before Answering: Self-Teaching LLMs to Learn Abstract Plans for Problem Solving"
_ICLR.cc/2025/Conference — ICLR 2025 Poster_

### Official Review · Reviewer_FT6B · 2024-10-19

**Soundness:** 3
**Presentation:** 3
**Contribution:** 2
**Rating:** 6
**Confidence:** 3

**Summary:**

The author present LEPA, a data generation pipeline. The data will not only contain the COT answer but also the planning process. The experiment shows that the LEPA beat other baselines in Hellaswag, Hendrycks MATH, BoolQ, and PIQA. The ablation study shows the importance of the self-peflection process and planning data in the data generation pipeline.

**Strengths:**

1. The paper is easy to follow.

2. The results show that planning data improves the performance of the LLMs.

3. The results in Figure 3 are impressive, the LEPA seems to continuously improve the performance even after 15 iterations.

**Weaknesses:**

1. Training planning data is not a new idea, paper like LUMOS[1] (specifically LUMOS-O) has already shown that planning data can improve the performance of the LLMs, although I admit that the LEPA is more simple than LUMOS-O.

2. There is only outcome judgment in data selection. Actually, LEPA doesn't judge the quality of the planning as well as the quality of the COT before the answer.

3. The comparison in Table 2 is a bit unfair. The LEPA's training data is (likely) bigger than the "Without Plan/Without Self-Reflection" settings if the settings are aligned with lines 349-360.


[1] LUMOS: LEARNING AGENTS WITH UNIFIED DATA, MODULAR DESIGN, AND OPEN-SOURCE LLMS

**Questions:**

1. In line 224, you claim that LEPA can "preventing the creation of false-positive data". However, even STaR can't prevent the creation of (step-level) false-positive data. It's even harder to judge whether the planning is correct/incorrect/meanless. So maybe you should give a evaluation standard to judge the quality of the planning data (and talking what is the defination of false-positive in planning).

2. Why not compare with other baselines mentioned in related work, especially those used MCTS?

3. Add some OOD benchmark (like MATH).

4. Discuss the reason why the planning data can improve the reasoning ability of the LLMs.

---

> ### Author Response · Authors · 2024-11-22
> **Response to Reviewer FT6B**
>
> Dear Reviewer:
>
> Thank you for the inspiring comments. We provide clarification to your questions and concerns as below. We appreciate any further questions or comments.
>
> **Q1:** Training planning data is not a new idea, paper like LUMOS[1] (specifically LUMOS-O) has already shown that planning data can improve the performance of the LLMs, although I admit that the LEPA is more simple than LUMOS-O.
>
> **A1:** There is a major difference between LUMOS and LEPA: LUMOS fine-tunes the LLM on a pre-collected dataset, while LEPA docuses on the problem of self-training, and finetunes on self-generated data. As discussed in Section 1 and the $ReST^{EM}$ paper [1], self-generated data can outperform pre-collected data, and addresses the thirst for high-quality data. The main contribution of LEPA is the idea of introducing and optimizing plans in the phase of data generation, which is drastically different from LUMOS's contribution. We have added this discussion in the refined paper.
>
>
> **Q2:** There is only outcome judgment in data selection. Actually, LEPA doesn't judge the quality of the planning as well as the quality of the COT before the answer.
>
> **A2:** Apart from final answer correctness, LEPA also judges planning quality & CoT quality in the self-reflection process. This analguous to optimizing plans and solutions from AI feedback, which helps LEPA to generate high-quality data.
>
> **Q3:** The comparison in Table 2 is a bit unfair. The LEPA's training data is (likely) bigger than the "Without Plan/Without Self-Reflection" settings if the settings are aligned with lines 349-360.
>
> **A3:** As claimed in line 266, for all experiments we run the algorithms until convergence, so all these variants are sufficiently trained.
>
> **Q4:** In line 224, you claim that LEPA can "preventing the creation of false-positive data". However, even STaR can't prevent the creation of (step-level) false-positive data. It's even harder to judge whether the planning is correct/incorrect/meanless. So maybe you should give a evaluation standard to judge the quality of the planning data (and talking what is the defination of false-positive in planning).
>
> **A4:** We apologize for the unclear presentation. The point of this claim is that LEPA prevents the problem of information bypassing, e.g., generating false-positive solutions due to diectly seeing the correct answer. We have modified this paragraph for a clearer presentation.
>
> **Q5:** Why not compare with other baselines mentioned in related work, especially those used MCTS?
>
> **A5:** We do not compare with these works because their contribution is orthogonal to ours. Our main contribution is the idea of introducing and optimizing anticiaptory plans in self-training's **data generation phase**, while these works focus on better-utilizing data in the **optimization phase**. As mentioned in Section 2.2, LEPA utilizes SFT for algorithm simplicity, and compares against SFT baselines for a fair comparison. LEPA can be easily applied to these advanced optimization algorithms without conflicts, and it is a promising future direction to integrate LEPA with these works to achieve better performance.
>
> **Q6:**  Add some OOD benchmark (like MATH).
>
> **A6:** We evaluated OOD performance by training on Hendrycks MATH and testing on the Math category of MMLU-Pro. As shown in Table 1 below, LEPA consistently outperforms baseline algorithms in this OOD setting.
>
>
> | CoT | Plan+CoT | $ReST$ | $ReST^{EM}$ | STaR | LEPA |
> | ------ | ------ | ------ | ------ | ------ | ------ |
> | 30.4% | 33.9% | 35.1% | 35.3% | 35.8% | **38.9%** |
>
> Table 1: OOD performance on the Math category of MMLU-Pro, with models trained on Hendrycks MATH. "CoT" and "Plan+CoT" refer to the initial LLM's performance with a zero-shot CoT prompt and the LEPA prompt, respectively.
>
> **Q7:** Discuss the reason why the planning data can improve the reasoning ability of the LLMs.
>
> **A7:** As discussed in Section 1 and Section 2.3, the planning data improve LLM reasoning ability in the following ways:
>
> 1) The plans serve as abstract blueprints that outlines the problem solving steps, avoiding the LLM from getting lost in the solution-generation process.
> 2) The plans are high-level generalizable knowledge that  empowers the LLM to solve similar problems more efficiently.
>
> References:
>
> [1] Singh A, Co-Reyes J D, Agarwal R, et al. Beyond human data: Scaling self-training for problem-solving with language models[J]. arXiv preprint arXiv:2312.06585, 2023.

---

> ### Comment · Reviewer_FT6B · 2024-11-22
> **Response by Reviewer FT6B**
>
> Thanks for authors' response. But I still have some confused point.
> Here is some of my point of view:
>
> 1. Algorithm 1, line 13 shows that you only judge the correctness of yi instead of pi. So A2 is incorrect. You can't find out the wrong plan pi if it gains a correct result yi. So LEPA still doesn't judge the quality of the planning as well as the quality of the COT before the answer.
>
> 2. About A4: can you use a color text in the pdf to emphasise the context you change?
>
> 3. Actually, data generation and data selection (better-utilizing data) can not be divided. LEPA also select data (only correct yi data will in the training set). I admit there is a gap between LEPA and MCTS method. Maybe you should consider MCTS as an "unrelated work" or compare it.
>
> 4. Your example can't prove your claim A7.1 (and I did not find any other evidence), since it is just like an improvement of the calculation, not coming from planning.
>
> 5.  Should "effectively" in  A7.2's claim be "efficiently"? And this claim looks like a circular argument to your main point.
>
> 6. It still seems just like a combination of  STaR[1] and LUMOS[2].
>
> [1] Star: Bootstrapping reasoning with reasoning.
>
> [2] LUMOS: LEARNING AGENTS WITH UNIFIED DATA, MODULAR DESIGN, AND OPEN-SOURCE LLMS

---

> > ### Author Response · Authors · 2024-11-23
> > **Response to Reviewer FT6B**
> >
> > Thank you for your feedback. Please find below the clarifications to your questions and concerns. If you have any further inquiries or comments, please feel free to post them, and we will be pleased to continue the discussion.
> >
> > **Q1:** LEPA still doesn't judge the quality of the planning as well as the quality of the COT before the answer.
> >
> > **A1:** As shown in Algorithm 1, line 11, LEPA first performs self-reflection on $p_i$ before judging answer correctness in line 13. This self-reflection serves as a quality judge & optimizer for planning, as it requires the LLM to find out whether there is any mistake in the plan, and modify the plan to improve it (as demonstrated by the detailed LEPA prompt in Figure 5 and Appendix A). We agree with the reviewer that LEPA does not obtain a step-level correctness judge for CoT, and may still generate data with incorrect CoTs and correct final answers. It is an important future direction to address this problem with methods such as learning step-level reward functions.
> >
> > **Q2:** Can you use a color text in the pdf to emphasise the context you change?
> >
> > **A2:** We have uploaded a new version of the paper that mark the changed context in red. Please check the pdf.
> >
> > **Q3:** Actually, data generation and data selection (better-utilizing data) can not be divided. Maybe you should consider MCTS as an "unrelated work" or compare it.
> >
> > **A3:** We agree with the reviewers. In the new version of the paper, we regard MCTS as unrelated work. As discussed in A1, it is an interesting future direction to incorporate both methods to achieve step-level CoT judgment in LEPA.
> >
> > **Q4:** Your example can't prove your claim A7.1 (and I did not find any other evidence), since it is just like an improvement of the calculation, not coming from planning.
> >
> > **A4:** The didactic example in Figure 1 shows that the plans serve as high-level abstract steps that guide actual solution generation. During solution generation, the LLM follows its plans, and avoids generating irrelevant steps such as dividing 45 by 2 in Figure 1(b). Note that Figure 1(b) also does not make any mistakes in calculation. It just gets lost in what to do at the next step.
> >
> > **Q5:**
> >
> > (1) Should "effectively" in A7.2's claim be "efficiently"?
> >
> > (2) This claim looks like a circular argument to your main point.
> >
> > **A5:**
> >
> > (1) Do you mean replacing "efficiently" with "effectively"? We agree with this point. Thanks for pointing out.
> >
> >
> > (2) To better understand this point, please refer to lines 225-229 in the pdf. The plans are generalizable as they are suitable for solving problems with similar underlying structures but different parameters. We also empirically demonstrate this generalizability power by testing on OOD benchmarks. Please refer to A6 in our original response.
> >
> > **Q6:** It still seems just like a combination of STaR and LUMOS.
> >
> > **A6:** As discussed in A1 in the original response,  the main contribution of our paper is the idea of introducing and optimizing plans in the phase of data generation. LEPA is not a combination of STaR and LUMOS, as it addresses the fundamental challenge of how to self-generate high-quality plans and corresponding solutions.

---

> > > ### Comment · Reviewer_FT6B · 2024-11-23
> > > **New Response by Reviewer FT6B**
> > >
> > > Thanks for the authors' response. I do like this idea. However, I am still concerned about its novelty, the quality of planning and COT, and the author's claim that planning data can improve reasoning ability. The author should figure out the proportion of each reason (gets lost, corrects calculation mistake, or other reasons) and the correctness of the planning and COT in its training set in both human eval (a small size) and GPT-4 (a large size). So, I will maintain my score now.

---

> ### Author Response · Authors · 2024-11-29
> **Response to Reviewer FT6B**
>
> Dear Reviewer:
>
> Thank you for your inspiring feedback. Please find below the clarifications to your concerns.
>
>
> **Q1:** Novelty.
>
> **A1:** The main novelty and contribution of LEPA is that it proposes the idea of jointly optimizing plan generation and solution generation in self-training. To our best knowledge, LEPA is the first work to propose integrating and optimizing plans in LLM self-training. Unlike LUMOS, in LEPA the planning and reasoning data needs to be automatically generated and optimized rather than pre-collected. LEPA addresses this fundamental challenge with a novel data generation pipeline.
>
> To further address the Reviewer's concern about LEPA's similarity to STaR+LUMOS, we would like to clarify that the main contribution of LEPA is not limited to SFT methods only, and is widely applicable to various self-training algorithms. To support this claim, we demonstrate a variant of LEPA that utilizes REINFORCE as the optimization algorithm in the model optimization phase. This variant is named LEPA+REINFORCE. The only difference from LEPA is that LEPA+REINFORCE does not discard failure data. Instead, it utilizes the solution correctness as the reward (1 for correct solutions, -1 for incorrect solutions). This implementation makes no modification to the data generation process. On Hendrycks MATH, LEPA+REINFORCE achieves a test accuracy of 30.6%, while LEPA achieves 30.2%. This performance improvement demonstrates the feasibility and effectiveness of incorporating LEPA with RL algorithms.
>
> **Q2:** Quality of planning and CoT. Whether planning data can improve reasoning ability. The author should figure out the proportion of each reason and the correctness of the planning and COT in its training set in both human eval and GPT-4.
>
> **A2:** To better demonstrate how planning data helps LLMs solve reasoning problems, we utilize GPT-4 to analyze algorithm test performance on Hendrycks MATH. The following 4 variants are evaluated: initial LLM with zero-shot CoT prompt (ZS CoT), initial LLM with LEPA prompt (Plan+ ZS CoT), LLM trained with ${ReST}$ ($ReST$), and LLM trained with LEPA (LEPA).
>
>
> We evaluate model performance on Hendrycks MATH's 5000 test problems, and prompt GPT-4 to classify the incorrect answers. For all variants, GPT-4 classifies their incorrect answers into 3 categories: 1. mistakes in problem-solving strategies (e.g., generating irrelevant steps like Figure 1); 2. mistakes in detailed calculation; and 3. other reasons. We manually checked 50 GPT-4 responses, and found that GPT-4 obtains a classification accuracy of 96% on these samples.
>
> As shown in Table 1 below, adding LEPA prompt to the initial model's prompt can reduce 40 calculation failures (1985 -> 1945) and 86 problem-solving strategies failures (1988 -> 1902). This demonstrates that adding a planning step can greatly reduce failures due to incorrect problem-solving strategies. As for self-training, LEPA makes significantly fewer problem-solving strategy failures than $ReST$ (1597 failures V.S. 1712 failures), which demonstrates that LEPA can indeed improve the model's planning ability.
>
>
> |  | ZS CoT | Plan+ ZS CoT | $ReST$ | LEPA |
> | ------ | ------ | ------ | ------ | ------ |
> | Mistakes in problem-solving strategies | 1988 | 1902 | 1712 | 1597 |
> | Mistakes in detailed calculation | 1985 | 1945 | 1851 | 1863 |
> | Other reasons | 51 | 46 | 25 | 29 |
> | Correct | 976 | 1107 | 1412 | 1511 |
>
> Table 1: Model evaluation on Hendrycks MATH. The first three rows demonstrate the number of different failure categories, and the last row demonstrates the number of correctly answered questions.
>
>
> We hope our responses adequately addresses your concerns. To further improve the paper, we will include all discussions in the rebuttal phase into the camera-ready version of the paper. If you have any further inquiries or comments, please feel free to post them, and we will be pleased to continue the discussion.
>
>
> References:
>
>
> [1] Zhang J, Kim J, O'Donoghue B, et al. Sample efficient reinforcement learning with REINFORCE[C]//Proceedings of the AAAI conference on artificial intelligence. 2021, 35(12): 10887-10895.

---

> > ### Comment · Reviewer_FT6B · 2024-11-29
> > **New Response by Reviewer FT6B**
> >
> > Thanks for the authors' response.
> >
> > For Q1, **your experiment does not align with your claim**, since it's obvious that reinforced learning can also be applied in STaR with your settings, so it is an advantage in STaR framework as used in LEPA. I should clarify that I do not deny that LEPA is better than STaR in some domains!
> >
> > For Q2, It does solve some of my concerns.
> >
> > I have raised my score accordingly.

---

> ### Author Response · Authors · 2024-11-30
> **Thank you for raising the score**
>
> We would like to thank the reviewer for raising the score! We also appreciate the valuable comments, which helped us significantly improve the paper's strengths.

---

### Official Review · Reviewer_5cwy · 2024-10-23

**Soundness:** 3
**Presentation:** 3
**Contribution:** 3
**Rating:** 8
**Confidence:** 4

**Summary:**

This paper introduces a novel training method LEPA for reasoning tasks. Unlike previous approaches that only focus on optimizing answer generation, LEPA also fine-tunes the LLM for plan generation. This is because planning represents a high-level abstract concept (meta-knowledge) that presents generalizable patterns across similar problems. LEPA prompts the LLM to generate an anticipatory plan and refine it until the plan can guide the LLM to a correct answer. After gathering the training data, LEPA uses SFT to optimize both the planning and answering generation. LEPA significantly enhances performance across various reasoning tasks.

**Strengths:**

1. The highlighted advantage is that the idea is simple and straightforward. It’s also quite clear why this would help performance, and the authors present it well with clear figures and examples. The method is easy to implement and can be easily generalized to other tasks, which makes this paper impactful.

2. The experiments are reasonable and effectively demonstrate the superiority of the method. There is sufficient analysis, including an analysis of inference computation.

**Weaknesses:**

1. My biggest concern is that this method may negatively affect the experience of using LLMs for simple problems. The current approach encourages the LLM to have careful planning before answering, which is unnecessary for simple problems and incurs unnecessary costs. In contrast, ReSF does not have this issue.

2. Secondly, I am concerned that this merely replaces the solution with an easily memorable method. I am curious whether it will provide sufficient improvement for few-shot scenarios, e.g., trained within a few MATH instances and tested with the others, or out-of-distribution reasoning problems, e.g., trained within the MATH dataset and tested with math questions in MMLU-pro.

3. More LLMs in experiments would be better. But this is not a big issue.

**Questions:**

1. In Table 1, the zero-shot CoT performance for llama3-8b-instruct is only 19%, which is a little bit too low. I acknowledge that the result would be different due to the prompts or something, but 19% is not very convincing to me. I would recommend you use other evaluation methods, e.g., simple-eval (https://github.com/openai/simple-evals) to re-evaluate the answer of the MATH dataset.

Also, see the comments on weaknesses. If issues are solved, I will improve my score.

---

> ### Author Response · Authors · 2024-11-22
> **Response to Reviewer 5cwy**
>
> Dear Reviewer:
>
> Thank you for the thoughtful comments. We provide clarification to your questions and concerns as below. We appreciate any further questions or comments.
>
> **Q1:**  This method may negatively affect the experience of using LLMs for simple problems.
>
> **A1:** We agree with the reviewers. Simple problems may not need planning, and training with LEPA can induce additional costs. Considering LEPA's effectiveness in solving complex benchmarks, a promising way to address this problem is to train the LLM only on hard problems that require planning, which can be filtered with human / AI feedbacks.
>
> **Q2:**  I am concerned that this merely replaces the solution with an easily memorable method. I am curious whether it will provide sufficient improvement for few-shot scenarios, e.g., trained within a few MATH instances and tested with the others, or out-of-distribution reasoning problems, e.g., trained within the MATH dataset and tested with math questions in MMLU-pro.
>
> **A2:** We evaluate OOD performance by training on Hendrycks MATH and testing on the Math category of MMLU-Pro. As shown in Table 1 below, LEPA consistently outperforms baseline algorithms in this OOD setting. This result demonstrates that the LEPA learns knowledge that is generalizable to OOD problems. We notice a performance gap between the initial LLM's performance to that reported in the MMLU-Pro paper. It is because 1) we use zero-shot CoT, and 2) we use the ChatCompletion method while the original paper uses the Completion method.
>
> | CoT | Plan+CoT | $ReST$ | $ReST^{EM}$ | STaR | LEPA |
> | ------ | ------ | ------ | ------ | ------ | ------ |
> | 30.4% | 33.9% | 35.1% | 35.3% | 35.8% | **38.9%** |
>
> Table 1: OOD performance on the Math category of MMLU-Pro, with models trained on Hendrycks MATH. "CoT" and "Plan+CoT" refer to the initial LLM's performance with a zero-shot CoT prompt and the LEPA prompt, respectively.
>
> **Q3:**  More LLMs in experiments would be better. But this is not a big issue.
>
> **A3:** We additionally evaluate algorithm performance on Llama 3.1 8B Instruct. As shown in Table 1 below, On the Hendrycks MATH dataset, the LEPA prompt can slightly improve over the zero-shot CoT prompt on the initial LLM. As for self-training, LEPA significantly outperforms the baseline algorithm $ReST$. These empirical results are consistent with our main results presented in Section 3.1.
>
> | CoT | Plan+CoT | $ReST$ | $ReST^{EM}$ | STaR | LEPA |
> | ------ | ------ | ------ | ------ | ------ | ------ |
> | 37.2% | 38.4% | 45.3% | 46.9% | 45.0% | **49.6%** |
>
> Table 1: Algorithm performance on Hendrycks MATH, with Llama 3.1 8B Instruct as the initial LLM. "CoT" and "Plan+CoT" refer to the initial LLM's performance with a zero-shot CoT prompt and the LEPA prompt, respectively
>
> **Q4:**  19% is not very convincing to me. I would recommend you use other evaluation methods, e.g., simple-eval to re-evaluate the answer of the MATH dataset.
>
> **A4:** We re-evaluate with simple-eval, and the results are demonstrated in Table 2 below. With the new evaluation, LEPA still outperforms baseline algorithms.
>
> | CoT | Plan+CoT | $ReST$ | $ReST^{EM}$ | STaR | LEPA |
> | ------ | ------ | ------ | ------ | ------ | ------ |
> | 26.1% | 28.5% | 31.2% | 31.4% | 29.2% |  **33.7%** |
>
> Table 2: Algorithm performance re-evaluated with simple-evals on Hendrycks MATH. "CoT" and "Plan+CoT" refer to the initial LLM's performance with a zero-shot CoT prompt and the LEPA prompt, respectively. LEPA consistently outperforms baseline algorithms.

---

> ### Author Response · Authors · 2024-11-25
> **Sincerely looking forward to further feedback**
>
> Dear Reviewer,
>
> Thank you for your time and efforts in reviewing our work. We have provided detailed clarification and experimental results to address the issues raised in your comments. If our response has addressed your concerns, we would be grateful if you could re-evaluate our work.
>
> If you have any additional questions or comments, we would be happy to have further discussions.
>
> Thanks,
>
> The authors

---

> > ### Comment · Reviewer_5cwy · 2024-11-26
> >
> > Thank you for the author's response, which addressed most of my concerns. I highly appreciate the generality and effectiveness of the proposed method. My main concern remains unresolved, i.e., ReSF might be a better solution in scenarios where planning is not required. I still maintain a positive view of the method and have adjusted my rating accordingly. I hope the authors can include further discussion and clarification on this point in subsequent versions of the PDF.

---

> ### Author Response · Authors · 2024-11-27
> **Thank you for raising the score**
>
> We would like to thank the reviewer for raising the score! We also appreciate the valuable comments, which helped us significantly improve the paper's strengths.
>
> We have added discussion on the point of "identifying whether a problem requires planning to solve" to Section 5 in the revised version of the paper. We will also include all other rebuttal discussions to the camera-ready version of the paper to further improve its quality.

---

### Official Review · Reviewer_VGrZ · 2024-10-30

**Soundness:** 2
**Presentation:** 2
**Contribution:** 2
**Rating:** 5
**Confidence:** 4

**Summary:**

The paper addresses the problem of what information should self-generated data encapsulate in the LLM self-training, by prompting an LLM (specifically Llama 3.1-8B-Instruct in this paper) to first generate an anticipatory plan before generating solutions to reasoning problems. Then optimize the anticipatory plan via self-reflection using the LLM itself until the plan leads to a correct solution. Once the corrected solution is generated, the reasoning problem and the plan as well as the solution are added to the training dataset to fine-tune the LLM.

Experimental results across four reasoning benchmarks show better performance compared to three baselines, especially after more training iterations.

**Strengths:**

1. The paper addresses an important problem that what essential information of self-generated data should an LLM learn during self-training, and considers an anticipatory plan as the essential information.
2. Experiment results show the the anticipatory plan can guide an LLM to generate correct solutions after the plan is refined iteratively through self-reflection and added to the training data for further fine-tuning.
3. The ablation studies validate the usefulness of key components of the proposed method.

**Weaknesses:**

1. Although the paper claims that the proposed method is also compatible with more sophisticated RL algorithms, the current version does not support the claim. If an RL algorithm is incorporated, the method will be quite different from the proposed one as the key components of current methods (e.g., self-reflection and fine-tuning method) should be significanly altered. So the performance of current method is limited to the LLM performance via prompting and self-reflection.
2. The justification of the anticipatory plan in Section 2.3 is not solid theorectically by simply referring to some work in cognitive science and the example shown in Figure 1.
3. The idea of generating a plan before solutions is similar to recent work on generating a plan using an LLM before actions for emobodied agents (e.g, [1][2]), however, the paper does not address the related work in Section 4. Also Section 4 lacks related work on self-reflection of LLMs (e.g, [3][4]), which is a key component of the proposeed method.

References:

[1] Wang, Z., Cai, S., Chen, G., Liu, A., Ma, X., and Liang, Y. Describe, explain, plan and select: interactive planning with llms enables open-world multi-task agents. In Thirty-seventh Conference on Neural Information Processing Systems, 2023.

[2] Yao, Shunyu, et al. "React: Synergizing reasoning and acting in language models." arXiv preprint arXiv:2210.03629, 2022.

[3] Madaan, A., Tandon, N., Gupta, P., Hallinan, S., Gao, L., Wiegreffe, S., Alon, U., Dziri, N., Prabhumoye, S., Yang, Y., et al. Self-refine: Iterative refinement with self-feedback. arXiv preprint arXiv:2303.17651, 2023.

[4] Shinn, N., Labash, B., and Gopinath, A. Reflexion: an autonomous agent with dynamic memory and self-reflection. arXiv preprint arXiv:2303.11366, 2023.

**Questions:**

1. Presentation of Table 3 is confusing and cannot figure it out in the title. Could you please explain the metrics in the Table and the relation of three variants of tokens to the three methods?
2. Have you ever tested the effectiveness of LEPA on other LLMs?
3. How do you think we can optimize the model when the solution to a reasoning problem is open-ended such that the evaluation of solution correctness is hard?

---

> ### Author Response · Authors · 2024-11-22
> **Response to Reviewer VGrZ**
>
> Dear Reviewer:
>
> Thank you very much for the feedback. We provide clarifications to your further questions as below. We appreciate any further questions or comments.
>
> **Q1:**  If an RL algorithm is incorporated, the method will be quite different from the proposed one as the key components of current methods (e.g., self-reflection and fine-tuning method) should be significantly altered. So the performance of current method is limited to the LLM performance via prompting and self-reflection.
>
> **A1:** We respectfully argue that incorporating RL needs no modification to the key components of LEPA, and does not affect our proposed idea of generating anticipatory plans before answering. The key difference between RL and SFT with self-generated data is RL's ability to utilize failure data, which improves learning efficiency.
>
> We demonstrate a simple implementation of incorporating LEPA with REINFORCE [1]. The only difference from LEPA is that LEPA+REINFORCE does not discard failure data. Instead, it utilizes the solution correctness as the reward (1 for correct solutions, -1 for incorrect solutions). This implementation makes no modification to the data generation process. On Hendrycks MATH, LEPA+REINFORCE achieves 30.6%, while LEPA achieves 30.2%. This performance improvement demonstrates the effectiveness of utilizing failure data with RL algorithms.
>
>
> **Q2:** The justification of the anticipatory plan in Section 2.3 is not solid theoretically by simply referring to some work in cognitive science and the example shown in Figure 1.
>
> **A2:** To our best knowledge, there is little theoretical justification for the effectiveness of intermediate step generation for LLM question answering, and this discussion is beyond the scope of our paper. Apart from the cognitive science view and the didactic example in Figure 1, we also justify the motivation of incorporating anticipatory plans from the view of meta-learning (Section 2.3), and the empirical results in Section 3.
>
>
>
> **Q3:** The idea of generating a plan before solutions is similar to recent work on generating a plan using an LLM before actions for embodied agents, however, the paper does not address the related work in Section 4. Also Section 4 lacks related work on self-reflection of LLMs, which is a key component of the proposed method.
>
> **A4:** Thank you for mentioning these related works. We have added related works on the plans and self-reflection in the refined version of the paper.
>
> **Q5:** Presentation of Table 3 is confusing and cannot figure it out in the title.
>
> **A5:** In Table 3 we demonstrate two metrics: "# of Tokens" is the average number of tokens in the LLM's response to test set problems. "Accuracy" is the model's accuracy on the test set. For the three variants: "Silence Tokens", "Correction" and "Long Solution", we train the LLM as described in Section 3.2, and the "# of Tokens" in Table 3 also represents the average number of tokens in the LLM's response to test set problems. We have added this clarification to the caption of the table.
>
> **Q6:** Have you ever tested the effectiveness of LEPA on other LLMs?
>
> **A6:** We additionally evaluate algorithm performance on Llama 3.1 8B Instruct. As shown in Table 1 below, On the Hendrycks MATH dataset, the LEPA prompt can slightly improve over the zero-shot CoT prompt on the initial LLM. As for self-training, LEPA significantly outperforms the baseline algorithm $ReST$. These empirical results are consistent with our main results presented in Section 3.1.
>
> | CoT | Plan+CoT | $ReST$ | $ReST^{EM}$ | STaR | LEPA |
> | ------ | ------ | ------ | ------ | ------ | ------ |
> | 37.2% | 38.4% | 45.3% | 46.9% | 45.0% | **49.6%** |
>
> Table 1: Algorithm performance on Hendrycks MATH, with Llama 3.1 8B Instruct as the initial LLM. "CoT" and "Plan+CoT" refer to the initial LLM's performance with a zero-shot CoT prompt and the LEPA prompt, respectively.
>
> **Q7:** How do you think we can optimize the model when the solution to a reasoning problem is open-ended such that the evaluation of solution correctness is hard?
>
> **A7:** For these problems that correctness is hard to judge, human / AI feedback is essential. A possible way is to learn a reward model to predict these feedbacks, and utilize LEPA + RL to maximize expected final return.
>
> References:
>
> [1] Zhang J, Kim J, O'Donoghue B, et al. Sample efficient reinforcement learning with REINFORCE[C]//Proceedings of the AAAI conference on artificial intelligence. 2021, 35(12): 10887-10895.

---

> > ### Comment · Reviewer_VGrZ · 2024-11-25
> >
> > Thanks for the authors' response. However, most of the explanations are still not convincing.
> > 1. The implementation of incorporating LEPA with REINFORCE is interesting. But is there still a need to incorporate self-reflection in this implementation, since we can utilize failure data for RL training? And this implementation seems to be quit different from that of proposed method in current version of the paper in terms of the combination of LEPA and REINFORCE. So if incorporating the new implementation into the paper, I think the novelty of the paper should be re-evaluated.
> > 2. If the theoretical justication of the anticipatory plan is beyond the scope of the paper, I doubt the value of introducing this theoretical explanation.
> > 3. How do you think the differences between generating a plan before solutions and generating a plan using an LLM before actions for embodied agents?
> > 4. The explanation of Table 3 is still confusing in terms of the connection of the three variants with the three method in the first row. Does a variant correspond to a method? If not, why put "Silence Tokens" below STaR and so on?
> > 5. Despite that the performance of Llama-3.1-8B-Instruct is consistenly better than that of Llama-3-8B-Instruct across methods, maybe testing another branch of LLMs can be better, such as Mistrial-7B-Instruct. But this is not important for evaluating the contribution of the paper.

---

> ### Author Response · Authors · 2024-11-25
> **Sincerely looking forward to further feedback**
>
> Dear Reviewer,
>
> Thank you for your time and efforts in reviewing our work. We have provided detailed clarification and experimental results to address the issues raised in your comments. If our response has addressed your concerns, we would be grateful if you could re-evaluate our work.
>
> If you have any additional questions or comments, we would be happy to have further discussions.
>
> Thanks,
>
> The authors

---

> ### Author Response · Authors · 2024-11-27
> **Further Response to Reviewer VGrZ**
>
> Dear Reviewer:
>
> Thank you for your inspiring feedback. Please find below the clarifications to your questions and concerns.
>
> **Q1:**
>
> (1) Is there still a need to incorporate self-reflection in this implementation, since we can utilize failure data for RL training?
>
> (2) And this implementation seems to be quite different from that of proposed method in current version of the paper in terms of the combination of LEPA and REINFORCE. So if incorporating the new implementation into the paper, I think the novelty of the paper should be re-evaluated.
>
> **A1:**
>
> (1) Yes, self-reflection is still beneficial to algorithm performance. It serves as a kind of in-context optimization that provides verbal evaluation on how to improve plan quality, as shown in Figure 4. This guidance is more detailed than RL's scalar evaluation, and helps LEPA to optimize plans more effectively. We test a variant of LEPA+REINFORCE that does not incorporate self-reflection. It achieves 29.6% on MATH, which underperforms both LEPA's 30.2% and LEPA+REINFORCE's 30.6%.
>
> (2) We have added the discussions and results of LEPA+REINFORCE to Section 3.2 in the new version of the paper. In the camera-ready version of the paper, we will evaluate LEPA+REINFORCE on all benchmarks, introduce LEPA+REINFORCE in the methods part (Section 2), and put the results in the main results part (Section 3.1).
>
>
>
> **Q2:** If the theoretical justication of the anticipatory plan is beyond the scope of the paper, I doubt the value of introducing this theoretical explanation.
>
> **A2:** Section 2.3 is a heuristic explanation of the anticipatory plans.  We believe the inspirations from cognitive science and meta learning, the didactic examples, as well as the empirical results are convincing and support our claim, and we notice that Reviewer 5cwy agrees that " It's also quite clear why this (the idea proposed by LEPA) would help performance, and the authors present it well with clear figures and examples."
>
> **Q3:** How do you think the differences between generating a plan before solutions and generating a plan using an LLM before actions for embodied agents?
>
> **A3:** They are in fact quite familiar, as they both try to solve reasoning problems with high-level abstraction. Previous works on planning for embodied agents focus on **utilizing** the LLM as a planner, while LEPA focuses on **optimizing** the LLM's planning ability. So it is also an interesting future direction to apply LEPA to embodied agent tasks.
>
> **Q4:** The explanation of Table 3 is still confusing in terms of the connection of the three variants with the three method in the first row. Does a variant correspond to a method?
>
> **A4:** No, there is no correspondence between algorithms in the same column. We put them in two rows due to the page width limit. We have added this clarification to the table caption in the new version of the paper.
>
> **Q5:** Despite that the performance of Llama-3.1-8B-Instruct is consistently better than that of Llama-3-8B-Instruct across methods, maybe testing another branch of LLMs can be better, such as Mistral-7B-Instruct. But this is not important for evaluating the contribution of the paper.
>
> **A5:** Thank you for the advice. Due to the computation resource limits, we are unable to provide results in the rebuttal period. We will add Mistral-7B-Instruct experiments in the camera-ready version of the paper. We will also include all the previous discussions in the rebuttal period to further improve the paper.

---

> > ### Comment · Reviewer_VGrZ · 2024-11-27
> >
> > Thanks for the authors' response. But I still have some concerns.
> > 1. The method gap between current paper and your promised final-version raises concerns about whether it should be the same paper to be evaluated. I'd like to leave the question to other reviewers and AC.
> > 2. I acknowledge the heuristic explanation of the anticipatory plans, but it is just heuristic rather than rigorous theoretical justification. So it is still not convincing.
> > 3. Previous works on planning for embodied agents also include optimizing the LLM's planning ability, such as utilizing different sources of feedback (e.g, environment feedback[1], other LLM critics' feedback[2], or human feedback[3]). So I doubt the novelty of current paper.
> >
> > References:
> > [1] Wang, Z., Cai, S., Chen, G., Liu, A., Ma, X., and Liang, Y. Describe, explain, plan and select: interactive planning with llms enables open-world multi-task agents. In Thirtyseventh Conference on Neural Information Processing Systems, 2023d.
> > [2] Shinn, N., Labash, B., and Gopinath, A. Reflexion: an autonomous agent with dynamic memory and self-reflection. arXiv preprint arXiv:2303.11366, 2023.
> > [3] Huang, W., Xia, F., Xiao, T., Chan, H., Liang, J., Florence, P., Zeng, A., Tompson, J., Mordatch, I., Chebotar, Y., et al. Inner monologue: Embodied reasoning through planning with language models. arXiv preprint arXiv:2207.05608, 2022.

---

> ### Author Response · Authors · 2024-11-27
> **Further Response to Reviewer VGrZ**
>
> Dear Reviewer:
>
> Thank you for your timely feedback. Please find below the clarifications to your further concerns.
>
> **Q1:** The method gap between current paper and your promised final-version raises concerns about whether it should be the same paper to be evaluated.
>
> **A1:** We believe that adding discussions about incorporating LEPA with RL does not affect the main claims and contributions of our paper. We welcome other Reviewers and AC to join this discussion.
>
> **Q2:** I acknowledge the heuristic explanation of the anticipatory plans, but it is just heuristic rather than rigorous theoretical justification. So it is still not convincing.
>
> **A2:** We believe that although LEPA does not obtain rigorous theoretical justifications, it still well supports its claims with sufficient motivations, didactic examples, and convincing empirical results, like many previous papers [1,2,3].
>
> **Q3:** Previous works on planning for embodied agents also include optimizing the LLM's planning ability. So I doubt the novelty of current paper.
>
> **A3:** The Reviewer may have misunderstood the position of our paper. These previous works optimize LLM planning abilities with **in-context optimization**, which does not involve optimizing the LLM's parameters. In contrast, LEPA focuses on improving LLM's planning and reasoning abilities with **fine-tuning**, which is orthogonal to and drastically different from the problem of LLM **in-context optimization**. In LEPA, self-reflection is only a method used for more efficient plan optimization, and is not LPEA's main contribution. To our best knowledge, LEPA is the first work to propose integrating and optimizing plans in LLM post-training fine-tuning.  So we believe LEPA is novel.
>
> References:
>
> [1] Wei, Jason, et al. "Chain-of-thought prompting elicits reasoning in large language models." Advances in neural information processing systems 35 (2022): 24824-24837.
>
> [2] Wang, Zihao, et al. "Describe, explain, plan and select: interactive planning with LLMs enables open-world multi-task agents." Advances in Neural Information Processing Systems 36 (2024).
>
> [3] Yang, Chengrun, et al. "Large language models as optimizers." The Twelfth International Conference on Learning Representations. 2024.

---

> > ### Comment · Reviewer_VGrZ · 2024-11-27
> >
> > Thanks for the authors' response. I have raised my score accordingly and leave Q1 to other reviewers and AC.

---

> > > ### Author Response · Authors · 2024-11-27
> > > **Thank you for raising the score**
> > >
> > > We would like to thank the reviewer for raising the score! We also appreciate your valuable comments, which inspire us to further improve the paper.

---

### Official Review · Reviewer_b2S7 · 2024-11-04

**Soundness:** 2
**Presentation:** 2
**Contribution:** 2
**Rating:** 6
**Confidence:** 4

**Summary:**

This paper introduces a self-training approach aimed at enhancing large language model (LLM) reasoning.
LEPA addresses the need for LLM self-generated data to contain meta-knowledge, rather than merely task-specific steps, enabling better generalization. By integrating anticipatory plans, LEPA provides a structured approach to problem-solving, reducing cognitive load, enhancing solution quality, and preventing the generation of incorrect rationales for correct answers.

**Strengths:**

1. Novel self-improvement method to improve the LLM's reasoning capability.

**Weaknesses:**

1. Only improve in-domain performance.
2. No OOD experiments.
3. Only one LLM (Llama 3 8B Instruct) is evaluated here.
4. Data generation pipeline is not clear illustrated.
5. Experiment section is not convincing with so few benchmarks.

**Questions:**

1. Will it work on other LLM?
2. How to deal with questions which have no correct answer after data creation pipeline?
3. How to better choose data?

---

> ### Author Response · Authors · 2024-11-22
> **Response to Reviewer b2S7 (1)**
>
> Dear Reviewer:
>
> Thank you very much for the feedback. We provide clarifications to your further questions as below. We appreciate any further questions or comments.
>
> **Q1:** Only improve in-domain performance. No OOD experiments.
>
> **A1:** We evaluate OOD performance by training on Hendrycks MATH and testing on the Math category of MMLU-Pro [1]. As shown in Table 1 below, LEPA consistently outperforms baseline algorithms in this OOD setting.
>
> | CoT | Plan+CoT | $ReST$ | $ReST^{EM}$ | STaR | LEPA |
> | ------ | ------ | ------ | ------ | ------ | ------ |
> | 30.4% | 33.9% | 35.1% | 35.3% | 35.8% | **38.9%** |
>
> Table 1: OOD performance on the Math category of MMLU-Pro, with models trained on Hendrycks MATH. "CoT" and "Plan+CoT" refer to the initial LLM's performance with a zero-shot CoT prompt and the LEPA prompt, respectively.
>
>
>
> **Q2:** Only one LLM is evaluated here. Will it work on other LLM?
>
> **A2:** We additionally evaluate algorithm performance on Llama 3.1 8B Instruct. As shown in Table 2 below, on the Hendrycks MATH dataset, the LEPA prompt can slightly improve over the zero-shot CoT prompt on the initial LLM. As for self-training, LEPA significantly outperforms the baseline algorithm $ReST$. These empirical results are consistent with our main results presented in Section 3.1.
>
> | CoT | Plan+CoT | $ReST$ | $ReST^{EM}$ | STaR | LEPA |
> | ------ | ------ | ------ | ------ | ------ | ------ |
> | 37.2% | 38.4% | 45.3% | 46.9% | 45.0% | **49.6%** |
>
> Table 1: Algorithm performance on Hendrycks MATH, with Llama 3.1 8B Instruct as the initial LLM. "CoT" and "Plan+CoT" refer to the initial LLM's performance with a zero-shot CoT prompt and the LEPA prompt, respectively
>
>
> **Q3:** Data generation pipeline is not clearly illustrated.
>
> **A3:** We demonstrate the data generation pipeline in the following parts: Figure 2 shows an outline of the data generation pipeline, Section 2.1 and Algorithm 1 present the pipeline in detail, and Appendix A describes the detailed prompts. We also notice that Both Reviewer 5cwy and Reviewer FT6B agree that the presentation is clear and easy to follow.  The following A4 and A5 address your detailed questions about the data generation pipeline. Please let us know if you have further questions about the pipeline.
>
> **Q4:** How to deal with questions which have no correct answer after data creation pipeline?
>
> **A4:** For the questions that are not correctly answered, LEPA discards these data, which is a common practice for self-training algorithms optimized with SFT. There are several possible future directions to extend LEPA to further utilize failure data:
>
> 1) Utilize success and failure data to train a success verifier, which can be utilized at either data generation [2] or inference time [3].
>
> 2) Integrate RL algorithms, which minimize the log-likelihood of failure data. We implement a variant of LEPA that utilizes REINFORCE [4] as the optimization algorithm. As it utilizes failure data during optimization, it achieves slightly better final performance than LEPA. On Hendrycks MATH, LEPA+REINFORCE achieves 30.6%, while LEPA achieves 30.2%.
>
> While these future directions are promising ways to better utilize data, these improvements focus on the model optimization phase, and are orthogonal to our main contribution of proposing the idea of incorporating anticipatory plans in LLM self-training.
>
>
> **Q5:** How to better choose data?
>
> **A5:** This is an important future direction. Indeed, choosing data by the final solution correctness may not be the most effective way. A promising future discussion is to incorporate human / AI feedback on data selection, and we believe this will further augment the benefits brought by LEPA's plan-before-answering data generation.
>
> **Q6:** Experiment section is not convincing with so few benchmarks.
>
> **A6:** We additional evaluated on CSQA and MMLU, and results are shown in Table 3 below.  LEPA consistently outperforms baseline algorithms on these benchmarks.
>
> | Benchmark | CoT | Plan+CoT | $ReST$ | $ReST^{EM}$ | STaR | LEPA |
> |-----------|-----|----------|------|-------------|------|------|
> | CSQA      | 67.1% | 69.3%    | 73.2% | 74.0%      | 74.1% | 75.2% |
> | MMLU      | 61.9% | 60.1%    | 64.3% | 65.6%       | 65.8% | 66.1% |
>
> Table 3: Algorithm performance on two additional benchmarks. "CoT" and "Plan+CoT" refer to the initial LLM's performance with a zero-shot CoT prompt and the LEPA prompt, respectively. LEPA consistently outperforms baseline algorithms on these benchmarks.

---

> > ### Comment · Reviewer_b2S7 · 2024-11-22
> >
> > Thank you for your detailed responses and comprehensive experiments. Your new exps have clarified my doubts and addressed my concerns. I have adjusted my score accordingly.

---

> ### Author Response · Authors · 2024-11-22
> **Response to Reviewer b2S7 (2)**
>
> References:
>
> [1] Wang, Yubo, et al. "Mmlu-pro: A more robust and challenging multi-task language understanding benchmark." arXiv preprint arXiv:2406.01574 (2024).
>
> [2] Hosseini A, Yuan X, Malkin N, et al. V-star: Training verifiers for self-taught reasoners[J]. arXiv preprint arXiv:2402.06457, 2024.
>
> [3] Zhang D, Zhoubian S, Hu Z, et al. ReST-mcts*: Llm self-training via process reward guided tree search[J]. arXiv preprint arXiv:2406.03816, 2024.
>
> [4] Zhang J, Kim J, O'Donoghue B, et al. Sample efficient reinforcement learning with REINFORCE[C]//Proceedings of the AAAI conference on artificial intelligence. 2021, 35(12): 10887-10895.

---

> ### Author Response · Authors · 2024-11-27
> **Thank you for raising the score**
>
> We would like to thank the reviewer for raising the score! We also appreciate the valuable comments, which helped us significantly improve the paper's strengths. We will add the rebuttal discussions to the camera-ready version of the paper to further improve its quality.

---

### Meta-Review · Area_Chair_sypD · 2024-12-24

**Metareview:**

This paper proposes LEPA for reasoning tasks. The main pipeline is that before really outputting the answer, it generates the plan of how to do the answer generation. The intuition behind why this works is that the plan has some high-level abstract concept that could host some general pattern across different problems with similar nature. In order to give LLM the ability to generate plan, it uses SFT to optimize the planning. All reviewers agree that this paper has good structure and easy to follow. Experiments show that the improvements are significant. Of course a more theoretical formulation to further justify this paper’s improvement in theory would be another huge plus for this paper. This paper is more empirical-driven. Because of this, reviewers have concern regarding this proposed idea’s relationship with other similar ideas. Furthermore, its performance benefit needs to be further justified with its computation cost. Yep, post-training scaling is more accepted now. I would encourage the paper to do more comprehensive analysis about cost/performance tradeoff, more analysis on whether it would fail, and more solid efforts to show its compatibility with RL (the improvements with REINFORCE seems week) to make this paper more solid.

**Additional Comments On Reviewer Discussion:**

The concern regarding references with other similar work is addressed in the rebuttal. Most of the reviewers' concern lie on more analysis about different cases that could arise in experiments, including its benefit with incorporation of RL. Those concerns have mainly been addressed and at least a reviewers increases scores accordingly by newly added experiments in the discussion.

---

### Decision · Program_Chairs · 2025-01-22

Accept (Poster)